# Research on Demand-Based Scheduling Scheme of Urban Low-Altitude Logistics UAVs

Honghai Zhang, Shixin Wu, Ouge Feng, Tian Tian, Yuting Huang and Gang Zhong *

College of Civil Aviation, Nanjing University of Aeronautics and Astronautics, Nanjing 211106, China
* Correspondence: zg1991@nuaa.edu.cn

**Abstract:** Aiming at the problem of the scheduling scheme of urban low-altitude logistics unmanned aerial vehicles (UAVs), this paper establishes a demand-based UAV scheduling scheme model using an improved simulated annealing algorithm, taking minimizing the cost of distribution as the objective function and considering restrictions such as UAV performance constraints, airspace constraints, and distribution constraints, among others. For verification, actual express data and airspace constraints in Shanghai are taken as examples. Two urban air traffic networks are constructed using road and building data. The analysis results show that the planning scheme of this model is superior to other forecasting models in terms of delivery cost and delivery time. In addition, this model can flexibly calculate the optimal scheduling scheme under the constraints of multiple parameters, according to the requirements of delivery volume, delivery distance, UAV performance, etc.

**Keywords:** air transportation; urban air mobility; simulated annealing algorithm; UAV logistics; scheduling scheme

## 1. Introduction

As a high-profile innovative industry, the current development of drones is in full swing. With the gradual maturity of technology, various types of drone applications have been further developed [1]. Although its application in the field of logistics has not yet entered a mature stage, many countries have carried out cutting-edge research and have also achieved many successful results. At the same time, both domestic and foreign logistics companies are trying to use small drones for distribution. For example, SF Express obtained the first Chinese drone aviation operation license in 2018, and Xunyi also launched drone delivery services in 2021. It can be seen that drone logistics has already become one of the emerging fields in drone applications [2]. Countries around the world have carried out research on drone logistics distribution, and using logistics drones is gradually becoming one of the ideal package delivery methods [3]. In the context of the post-epidemic era, drone logistics gives full play to the advantages of contactless transportation. Obviously, it is bound to have a place in the future logistics field. Under this development background, relevant research on the scheduling scheme of logistics drones is necessary. An effective scheduling scheme will be conducive to the rational planning and construction of logistics infrastructure and also to the improvement in logistics transportation systems, which are of great significance to improving logistics transportation efficiency and reducing logistics costs.

There have been many studies on logistics drones both in China and abroad [4]. On the demand forecasting of logistics drones, Lakshmi and other foreign scholars built a linear programming model to predict the demand for logistics drones for the purpose of maximizing the operator's income [5]. Marc et al. used methods of data analysis and comparative analysis to predict the demand for logistics drones in the United States in 2050 [6]. Doole et al., considering factors such as the transportation capacity of drones and the proportion of urban population, gradually determined the number of packages transported by drones from the perspective of pessimistic, optimistic, and practical probability

values and then determined the demand for logistics drones [7]. Chauhan et al. proposed a robust solution with regard to the uncertainty in battery availability and consumption [8]. Glick et al. developed a modeling framework to analyze drone delivery reliability under stochastic demand and meteorological conditions [9]. Dell'Amico M et al. considered the traveling salesman problem, in which trucks and drones cooperate to deliver packages to customers [10]. Wu Y et al. further reduced flight delays by optimizing the sequence of drones performing conflict resolution [11]. Eslamipoor R et al. propose a two-level, multi-product, and multi-cycle integrated inventory-transportation planning model for carbon emissions [12] and another model for locating product collection centers considering risk and environmental factors [13]. She R et al. proposed a finite element scheme to numerically solve the flow equilibrium and calculate the system performance and investigated two specific test scenarios for last-mile freight delivery systems [14]. At present, there is relatively little research on drone demand in China. Based on the strategic background of civil-military integration, Chen Gang et al. considered the heterogeneity of demand points and distribution centers as well as the vulnerability of the network when investigating the location problem of drone distribution centers. Aiming at the different needs of different types of demand points for distribution centers, they established a drone distribution center model with the minimum total network mileage as the objective function [15]. Zhou Lang proposed a distribution mode of "vehicle + drone" for rural e-commerce logistics. In order to solve the problem of its path optimization, he built a variety of distribution-path optimization models, using genetic algorithms and other algorithms to solve the models [16]. Zhang Fang established a multi-stage drone-demand forecasting model with the objective function of maximizing express transportation volume and minimizing safety costs [17].

In terms of research on logistics node location model algorithms, there are mainly two categories: precise algorithms and heuristic algorithms. An accurate algorithm refers to a method that can directly calculate the optimal solution of a model. The more commonly accurate algorithms include the branch-and-bound method, the enumeration method, and the dynamic programming method, etc. Among them, the branch-and-bound method is the most commonly used method by scholars. Efroymson [18] used the branch-and-bound method to solve the problem of node location without capacity constraints. Khumawala [19] used an improved principle combined with the branch-and-bound method to calculate the location problem without capacity constraints based on previous studies. Davis [20] studied the location problem of logistics facilities with capacity constraints using precise algorithms. Heuristic algorithms can relatively optimize the solving steps and also shorten the solving time, so many scholars use heuristic algorithms to solve models and obtain satisfactory solutions. Heuristic algorithms include the simulated annealing algorithm, the particle-swarm optimization algorithm, the Lagrange algorithm, and the genetic algorithm. Antunes and Peeters [21] used a simulated annealing algorithm to solve node-layout problems related to multiple time periods. Jayaraman and Ross [22] used a simulated annealing algorithm to calculate the logistics network-layout model in multi-stage situations. The idea of particle-swarm optimization is similar to that of the simulated annealing algorithm. It is to set a random initial solution and then, through multiple iterations and fitness evaluation, track the currently searched optimal solution to find the global optimal solution of the model. Hu Wei [23] designed an improved particle-swarm optimization algorithm to solve the location model of distribution centers and demonstrated through examples that the improved particle-swarm optimization algorithm can significantly improve the model's solving speed and accuracy.

In terms of logistics-drone-scheduling research, Kim M et al. [24] combined logistics drones with public transportation systems to form a heterogeneous multi-agent system. By solving the vehicle routing problem (VRP), they found a path for each package, thus finding a path for cost optimization under the conditions of a given heterogeneous multi-agent system and minimizing the number of drones required for delivery. Sigala A et al. [25] used small autonomous drone systems (UAS) located below 500 feet in large urban areas as the analysis object and used data analysis and comparative analysis methods to predict

the demand for logistics drones in the United States in 2050 and the economic benefits of using drones for package delivery. Vempati L et al. [5] took PrimeAir in Amazon, Phoenix, Arizona, as an example to give a linear programming and simulation model for transporting packages. In this paper, two models are established. The first is a linear programming model that aims to maximize the revenue of operators and predicts the demand for logistics drones on this basis. The second is a simulation model that describes the impact of changes in consumer demand on logistics drone distribution. The simulation model is based on the output of the linear programming model. Grippa et al. [26] studied the lower bound of delivery time and infrastructure expenditure for goods transported from warehouses to customers through drones and proposed a strategy to minimize the overall workload. This strategy can be expanded based on the number of warehouses and drones, with optimal performance at low loads and normal operation at high loads. The simulation shows that if the number of drones in each time period is sufficient, it can meet the work needs of any load.

To sum up, part of the studies only conducted research on demand forecasting methods and failed to reflect the application of drones as an emerging mode of transportation in logistics. Some studies only partially covered the requirements of drones and did not regard specific requirements as necessary constraints, while others only considered a single element, which is difficult to apply to actual delivery scenarios. In other words, the existing studies fail to take into account the performance of drones and the distribution environment of urban low-altitude logistics, which makes the demand-based scheduling results lack certain credibility.

In order to enrich the research content of drone scheduling schemes, this paper learns from the ideas of the literature [17,27], considers drone performance constraints such as load and endurance, and also considers the constraints of airspace conditions to establish a simulation model [28] for drone demand-based scheduling schemes. Based on the characteristics of the model, an improved simulated annealing algorithm is used to solve the scheduling scheme of urban low altitude logistics unmanned aerial vehicles for two types of route networks. In summary, this paper not only establishes a multi-objective optimization model on the basis of considering various constraints such as drone performance and environmental restrictions but also conducts a parameter-adjustment analysis on the model, which increases the innovation of the article and the feasibility of the model and can also provide some reference for the future development of logistics drones [29,30].

## 2. Problem Description and Modeling

### 2.1. Problem Description

Considering airspace constraints and the low-altitude operation conditions of the drone itself, this paper predicts drone demand and the corresponding costs of delivering express packages from distribution points to various demand points for two urban air traffic networks constructed using road and building data. In order to make the scheduling results meet the requirements of the logistics development trend, the following problems need to be solved:

(1) Forecast the delivery cost required to complete all delivery tasks under the optimal situation;
(2) In the case of the minimum distribution cost, predict the ideal delivery route of the distribution point under the optimal situation.

### 2.2. Model Assumptions

According to the "Notice on Soliciting Opinions on the Development Roadmap V1.0 of Civil Unmanned Aerial Navigation" issued by the Civil Aviation Administration of China [31] and the Provisional Regulations on Flight Management of Unmanned Aerial Vehicles (Draft) approved by the State Council [32] issued by the Central People's Government of the People's Republic of China, the following assumptions are made:

(1) Generally, the dimensions of the packages sent by mail are not smaller than 20 × 20 cm (length × width); length, width, and height shall not exceed 2.5 m, 1.5 m, and 1.5 m,

respectively; packages are generally wrapped in a square or rectangular shape (it is assumed that the influence of the actual shape of the cargo is not considered when the drones transport the cargo);

(2) It is assumed that the working time of the logistics drones is not affected by the weight of the loaded cargo;

(3) It is assumed that the drones keep flying at a constant speed during the delivery process;

(4) It is assumed that each drone initially carries a fully charged set of batteries during the delivery process, which powers the drone's flight.

*2.3. Model Building*

2.3.1. Constraint Conditions

1. Drone performance constraints:

(1) Flight range

The distance $l_{ij}$ from the delivery point $i$ to the demand point $j$ should meet the longest flight range $L_{ij}^m$ of the drone $m$ [27], which is expressed as:

$$l_{ij} \leq L_{\max}^m \quad \exists m \in M \tag{1}$$

(2) Load

Only when the weight $w_k$ of the package $k$ is within the maximum load limit $W_{ij}^m$ of the drone $m$, can the package be transported by the drone $m$ [29], which can be expressed as:

$$w_k \leq W_{\max}^m \quad \exists m \in M \tag{2}$$

(3) Working hours

The working time $t_{ij}^m$ of the drone should not exceed its own working time limit $T^m$. If the working time is too long, its battery pack will be exhausted, and the drone will stop working. The formula is expressed as:

$$\sum_{j=1}^{J} t_{ij}^m \leq T^m \quad \exists m \in M \tag{3}$$

(4) Charging duration

When the drone $m$ stops working due to insufficient power, the battery pack needs to be charged, and the drone $m$ must wait until the charging is completed before continuing to work. The formula is expressed as:

$$x_i = \begin{cases} 0 \ \text{UAV } i \text{ charging} \\ 1 \ \text{UAV } i \text{ not charging} \end{cases} \tag{4}$$

(5) Backup battery

Consider that when the power of the drone $m$ is insufficient, a backup battery $n$ can be selected from the total backup battery pack $N$, and then the primary battery pack will be charged. The drone $m$ can continue to work after replacing the primary battery, but if the backup battery pack is charging and there is no spare battery available, the drone $m$ will stop working and must be charged. The formula is expressed as:

$$y_i = \begin{cases} 0 \ \text{Backup battery } n \text{ available} \\ 1 \ \text{Backup battery } n \text{ not available} \end{cases} \tag{5}$$

2. Airspace constraints:

(1) Flight altitude

As the drone $m$ flies in the low-altitude airspace, the flight altitude $h^m$ should meet the maximum flight altitude limit $H_{max}$ and the minimum limit $H_{min}$ allowed in the airspace. This is expressed by the formula:

$$H_{min} \leq h^m \leq H_{max} \quad \forall m \in M \tag{6}$$

(2)  Flight speed

The drone $m$ flies in the low-altitude airspace, and its flight speed $v^m$ must meet the maximum and minimum allowable flight speeds $V_{max}$ and $V_{min}$ in the airspace. In this article, we stipulate that the drone has a speed of $V_{min}$ when loading goods for distribution and a speed of $V_{max}$ when returning empty. This is expressed as:

$$V_{min} \leq v^m \leq V_{max} \quad \forall m \in M \tag{7}$$

2.3.2. Objective Functions

(1)  Time cost

This paper assumes that the time-cost coefficient $T_\alpha$ is a periodic function, and the period is denoted as $T_t$. Considering the impact of peak times, $T_\alpha$ can be approximately expressed as an exponential function:

$$T_\alpha = \begin{cases} T_{\alpha 0} & t_{ij}^m \leq t_1 \\ k_1 \cdot e^{t_{ij}^m} + b_1 & t_1 < t_{ij}^m <= t_2 \\ k_2 \cdot e^{t_{ij}^m} + b_2 & t_2 < t_{ij}^m <= t_3 \end{cases} \tag{8}$$

where $T_{\alpha 0}$ represents the time-cost coefficient under standard conditions; other variables represent the time-cost coefficient during peak times.

Take the time-cost of drone transportation as a factor, which can be expressed as:

$$Z_1 = \sum_{m=1}^{M} \sum_{i=1}^{A} \sum_{j=1}^{J} t_{ij}^m \cdot T_\alpha \tag{9}$$

where $t_{ij}^m$ refers to the delivery time of the drone, $T_\alpha$ represents the time-cost coefficient of the delivery, and $Z_1$ represents the total time cost of completing all the delivery tasks. $A$ represents the set of routes for all drone flights, and obviously, the distance of the drone flight route is greater than the total distance of the entire route itself. $J$ is a collection of requirement points. $Z_1$ represents the time spent. This is the total cost incurred by each drone during each delivery.

(2)  Risk cost

Let $D_\beta$ denote the risk-cost coefficient of the drone transportation process, which is related to the transportation distance and peak times.

On the one hand, the drone risk-cost coefficient is related to the distribution distance. The longer the distribution distance, the more likely it is that the risk will occur. Therefore, the risk-cost coefficient increases with the distribution distance. Let $L_t$ indicate the total number of routes in the network, then:

$$l_a = \frac{\sum_{i=1}^{A} \sum_{j=1}^{J} l_{ij}}{L_t} \tag{10}$$

where $l_a$ represents the average distance of each delivery route.

Let $c_a$ represent the risk-cost coefficient corresponding to the distance $l_a$, then the drone risk-cost coefficient of each delivery route is:

$$D_\beta{}^{ij} = \frac{l_{ij}}{l_a} c_a \tag{11}$$

On the other hand, the drone risk-cost coefficient is related to peak times. During peak times of busy distribution, the risk-cost coefficient also increases due to the presence of vehicles, pedestrians, and other influencing factors. This paper assumes that the risk-cost coefficient $D_\beta$ is a periodic function, and the period is recorded as $T_t$. Therefore:

$$D_\beta = \begin{cases} D_\beta{}^{ij} & t_{ij}^m \le t_1 \\ k_3 \cdot e^{t_{ij}^m} + b_3 & t_1 < t_{ij}^m <= t_2 \\ k_4 \cdot e^{t_{ij}^m} + b_4 & t_2 < t_{ij}^m <= t_3 \end{cases} \tag{12}$$

where $t_1$, $t_2$, $t_3$ are the same as those in Formula (8).

Take the risk cost of drone transportation as a factor, which can be expressed as:

$$Z_2 = \sum_{m=1}^{M} \sum_{i=1}^{A} \sum_{j=1}^{J} x_i \cdot y_i \cdot l_{ij}^m \cdot D_\beta \tag{13}$$

where $l_{ij}^m$ is the delivery distance of the drone, $D_\beta$ is the risk-cost coefficient of the delivery, and $Z_2$ represents the total risk cost of completing all the delivery tasks. The farther the distance, the closer the time is to peak hours and the higher the risk.

(3) Maneuvering cost

Use $D_\delta$ to represent the maneuvering-cost coefficient of the drone transportation process. Every time the drone $m$ is dispatched to perform the delivery task, a maneuvering cost will be generated. The maneuvering-cost coefficient is related to the load capacity of the drone. The higher the load capacity of the drone, the greater the maneuvering-cost coefficient. The formula for calculating the maneuvering-cost coefficient is as follows:

$$D_\delta = \begin{cases} c_1 & w_k = w_1 \\ c_2 & w_k = w_2 \\ \dots \\ c_n & w_k = w_n \end{cases} \tag{14}$$

Take the maneuvering cost of drone transportation as a factor, which can be expressed as:

$$Z_3 = \sum_{m=1}^{M} F_m \cdot D_\delta \tag{15}$$

where $F_m$ is the maneuvering time of the delivery, $D_\delta$ is the maneuvering-cost coefficient during the delivery, and $Z_3$ represents the total maneuvering cost when all the delivery tasks are completed. The farther the distance, the closer the time is, indicating the cost of maneuvering. This means that every drone deployment incurs costs. During peak hours, the risk is higher.

2.3.3. Building the Model

In summary, the delivery cost $Z$ is composed of the time cost $Z_1$, the risk cost $Z_2$, and the maneuvering cost $Z_3$, the scheduling scheme model of urban low-altitude logistics drones can be described as follows:

$$\min Z = Z_1 + Z_2 + Z_3 \tag{16}$$

$$s.t. \begin{cases} l_{ij} \leq L_{\max}^m & \exists m \in M \\ w_k \leq W_{\max}^m & \exists m \in M \\ \sum\limits_{j=1}^{J} t_{ij}^m \leq T^m & \exists m \in M \\ x_i = \begin{cases} 0 \ \text{UAV } i \ \text{charging} \\ 1 \ \text{UAV } i \ \text{not charging} \end{cases} \\ y_i = \begin{cases} 0 \ \text{Backup battery } n \ \text{available} \\ 1 \ \text{Backup battery } n \ \text{not available} \end{cases} \\ H_{\min} \leq h^m \leq H_{\max} & \forall m \in M \\ V_{\min} \leq v^m \leq V_{\max} & \forall m \in M \end{cases} \quad (17)$$

## 3. Algorithm

This paper uses an improved simulated annealing algorithm to solve the model. Specifically, dynamic allocation is added on the basis of the standard simulated annealing algorithm. For the dynamic allocation algorithm, Ref. [17] is referred to. Based on the literature, the process has been properly improved. The dynamic allocation algorithm can predict the best delivery scheme and drone demand dispatches. The simulated annealing algorithm is used to iteratively update the scheme to obtain the optimal one. The combination of the two can solve the problem of the variability of model parameters, so that the parameters can be dynamically adjusted during the calculations to find the best solution. The overall algorithm flow is shown in Figure 1.

The key point of the improved simulated annealing algorithm in this paper is to add a dynamic allocation mechanism to the standard simulated annealing process to improve the efficiency of the algorithm. The specific steps are as follows:

Step **1**: Initialize the parameters. Make the initial temperature $T$ sufficiently high and generate feasible routes for the drones to obtain the initial solution $S$. The number of iterations for each $T$ value is $L$.

Step **2**: Repeat steps 3 to 10 for $k = 1, \cdots, L$.

Step **3**: Start the dynamic allocation of the drones.

Step **4**: Obtain the package quantity $N_s^e$ at the demand point. If it is greater than zero, find out the optimal delivery scheme of the drone and go to Step 2; otherwise, go to Step 3.

Step **5**: Judge whether the power of the drone is sufficient for the delivery. If the power is not enough, replace the primary battery with the backup battery and charge the primary battery. If the backup battery is also insufficient, the drone will be charged. Go to Step 7.

Step **6**: Judge whether all the demand points have been processed. If they have all been processed, the algorithm ends. A feasible solution will be generated, that is, the new solution $S'$ will be obtained. Otherwise, the next demand point will be processed, and Step 4 will be entered.

Step **7**: Judge whether the working time of the drone is sufficient. If it is not enough, assign a new drone and go to Step 4. If it is sufficient, update the package volume at the demand point to $N_s^e = N_s^e - n$ and go to Step 4.

Step **8**: Calculate the increment $\Delta T = C(S') - C(S)$, where $C(S)$ is the evaluation function.

Step **9**: If $\Delta T < 0$, accept $S'$ as the new current solution; otherwise, accept $S'$ as the new current solution with probability $\exp(-\Delta T/T)$.

Step **10**: If the termination conditions are met, the current solution is output as the optimal solution, and the program ends.

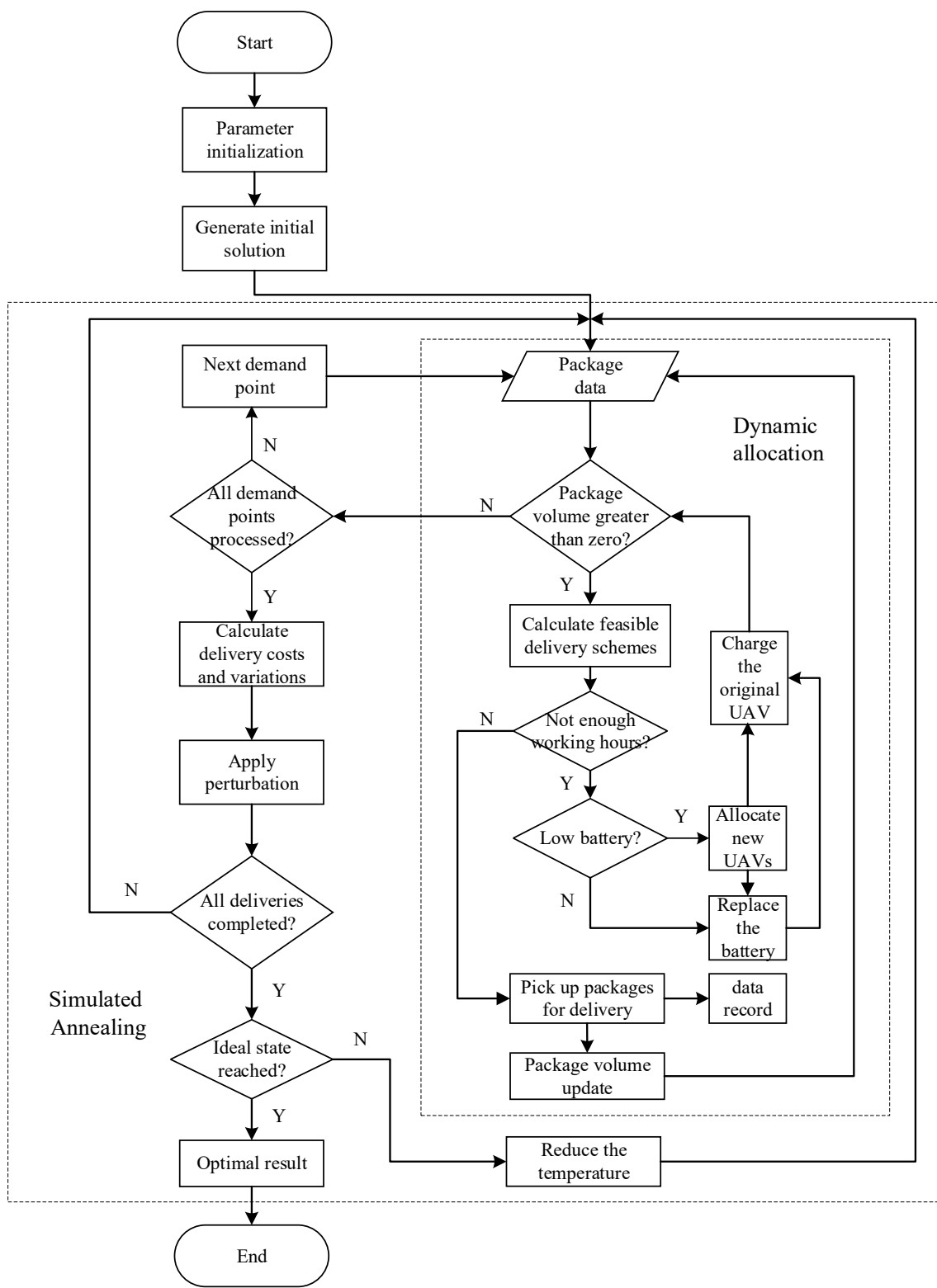

**Figure 1.** Solution process.

## 4. Example Verification and Analysis

### 4.1. Sample Data and Parameter Settings

To verify the effectiveness of the model and algorithm in this paper, the actual data of express business volume, distribution points, and demand points in Shanghai are selected as sample data, as shown in Table 1. Select a real distribution point in the region to analyze

the business volume for a certain day. The distribution point has 98 demand points and 5760 packages. According to the regulations on low-altitude use issued by the region, suitable delivery routes for drones have been planned in the preliminary work, that is, two urban air traffic networks constructed using road and building data. When planning the two-layer route, the A* algorithm and 500 × 500 grid network are used to avoid the airspace no-fly zone and ground obstacles. The flow chart is shown in Figure 2. The urban air traffic network (RN) constructed using road data is shown in Figure 3, and the urban air traffic network (BN) constructed using building data is shown in Figure 4. The routes on the two floors represent a 100 m high departure route and a 150 m high return route, respectively. The initial delivery time is set at 8:00 a.m. According to constraints (8) and (11), the peak time will start 3 h later and end 5 h later. In order to make the flight of drones during transportation as real as possible, this paper sets the simulation according to the performance parameters of drones currently used in logistics transportation on the market, as shown in Tables 2 and 3. In this article, the idealized situation is considered, and it is stipulated that the drone equipment used in the experiment has been obtained without incurring any other costs due to the purchase of equipment.

**Table 1.** Sample data for express demand forecasting.

| Demand Point Serial Number | Packages to Be Delivered (kg) |
|---|---|
| B0089 | 68 |
| B0189 | 50 |
| B0268 | 80 |
| B0324 | 69 |
| B0427 | 77 |
| … | … |
| B8796 | 58 |
| B9078 | 9 |
| B9111 | 79 |

**Table 2.** Initial model simulation data.

| Parameter | Value | Parameter | Value |
|---|---|---|---|
| Maximum cargo capacity $W_{\max}^m/\mathrm{kg}$ | 20 | Risk cost of road network per unit distance $D_{\beta1}{}^{ij}/\mathrm{Yuan}\cdot\mathrm{km}^{-1}$ | 0.4 |
| Maximum flight range $L_{\max}^m/\mathrm{km}$ | 18 | Risk cost of building network per unit distance $D_{\beta2}{}^{ij}/\mathrm{Yuan}\cdot\mathrm{km}^{-1}$ | 0.2 |
| Minimum flight speed $v_{\min}/\mathrm{m}\cdot\mathrm{s}^{-1}$ | 1 | Drone maneuvering cost $D_\delta/\mathrm{Yuan}\cdot\mathrm{sorties}^{-1}$ | 0.5 |
| Maximum flight speed $v_{\max}/\mathrm{m}\cdot\mathrm{s}^{-1}$ | 10 | Number of drones $I$ | 50 |
| Maximum flight altitude limit $H_{\min}^m/\mathrm{m}$ | 100 | Number of backup batteries $N$ | 20 |
| Minimum flight altitude limit $H_{\max}^m/\mathrm{m}$ | 150 | Charging duration of drone $h$ | 2 |
| Standard time-cost coefficient $T_{\alpha0}/\mathrm{Yuan}\cdot\mathrm{min}^{-1}$ | 0.1 | Working hours of drone $T_m/\mathrm{min}$ | 30 |

**Table 3.** Simulation data for coefficients.

| Parameter | Value | Parameter | Value |
|---|---|---|---|
| $k_1$ | $\dfrac{T_{\alpha0}}{e^4-e^3}$ | $b_1$ | $\dfrac{T_{\alpha0}(e^4-2e^3)}{e^4-e^3}$ |
| $k_2$ | $\dfrac{T_{\alpha0}}{e^4-e^5}$ | $b_2$ | $\dfrac{T_{\alpha0}(e^4-2e^5)}{e^4-e^5}$ |
| $k_3$ | $\dfrac{D_\beta{}^{ij}}{e^4-e^3}$ | $b_3$ | $\dfrac{D_\beta{}^{ij}(e^4-2e^3)}{e^4-e^3}$ |
| $k_4$ | $\dfrac{D_\beta{}^{ij}}{e^4-e^5}$ | $b_4$ | $\dfrac{D_\beta{}^{ij}(e^4-2e^5)}{e^4-e^5}$ |

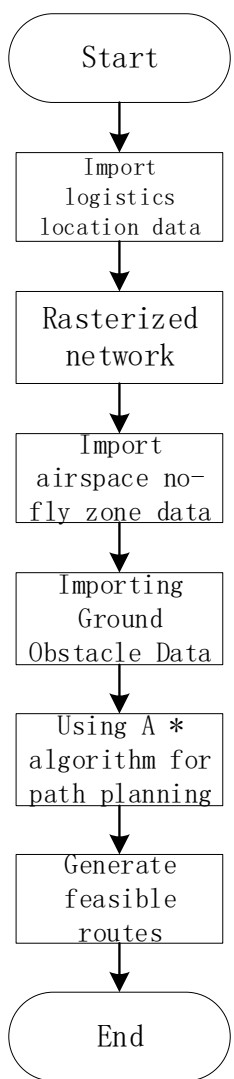

**Figure 2.** Flow Chart of Urban Air Traffic Network Path Planning.

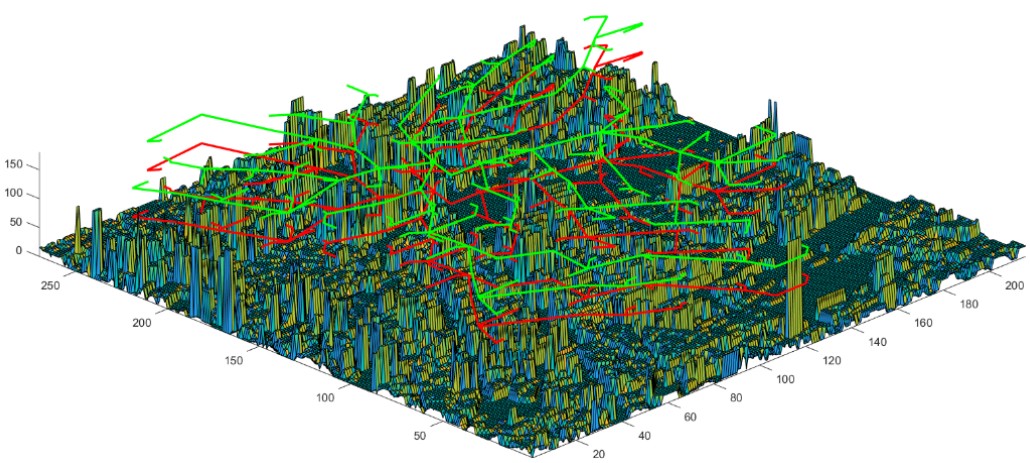

**Figure 3.** Urban air traffic network constructed by road data.

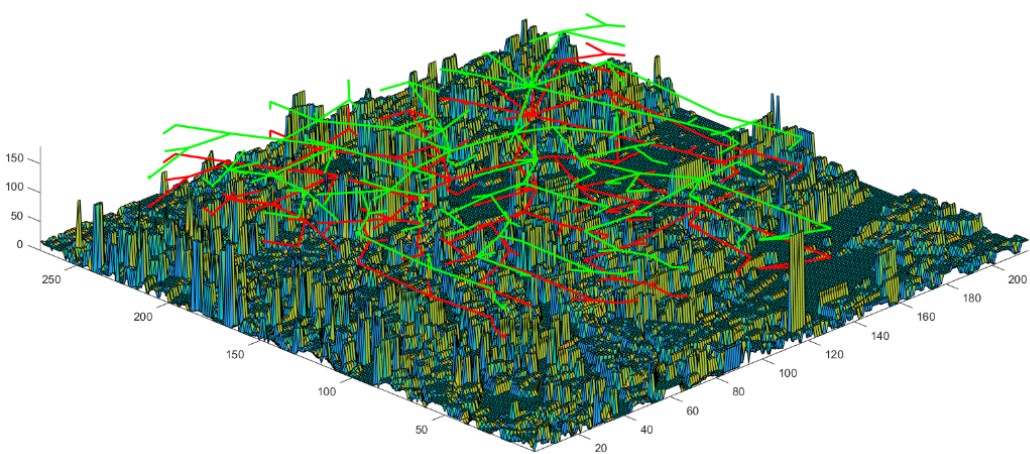

**Figure 4.** Urban air traffic network constructed by building data.

### 4.2. Analysis of the Calculation Results

According to the algorithm proposed in this paper, the initial temperature of simulated annealing is set to 100, the lowest temperature is 0.03, inverse cooling is adopted, and the parameter is set to 0.001. Under the premise that each parameter is set and the distribution airspace environment remains unchanged, the future drone scheduling scheme for this distribution point is calculated using Python programming, and the predicted results and drone allocation are shown in Table 4. The iterative effect of the two routes based on the improved simulated annealing algorithm is shown in Figure 5. The flight schedules of the drones on both routes are shown in Tables 5 and 6.

**Table 4.** Results of unmanned aerial vehicle scheduling schemes under different routes.

|  | Average Charging Times of Drone | Average Use Times of Backup Battery | Time Cost | Risk Cost | Maneuvering Cost | Delivery Cost | Delivery Time |
|---|---|---|---|---|---|---|---|
| RN | 2 | 3 | 474.31 | 1326.80 | 288.0 | 2089.11 | 5 h 36 min 33 s |
| BN | 2 | 3 | 621.67 | 716.70 | 288.0 | 1626.37 | 5 h 47 min 44 s |

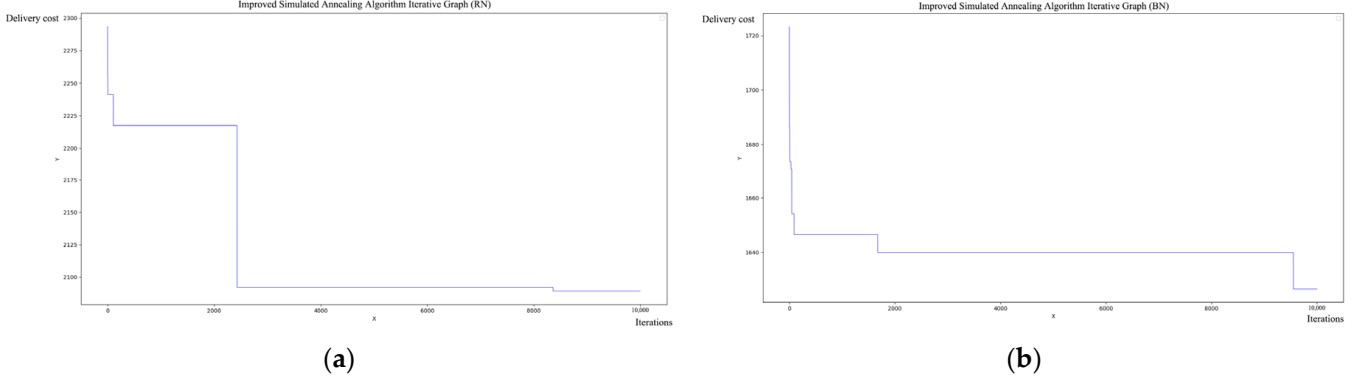

(**a**)　　　　　　　　　　　　　　　　　　　　　　　　　　　(**b**)

**Figure 5.** Improved Simulated Annealing Algorithm Iterative Graph (RN and BN), where (**a**) represents the RN route and (**b**) represents the BN route.

It can be seen from Table 4 that, under the two different routes, the average charging times of the drone and the average use times of the backup battery are the same. Although the BN route takes longer, it also greatly reduces the risk cost, and ultimately, the delivery cost of the BN route is much lower.

**Table 5.** Drone Dispatch Flight Schedule (RN).

| Delivery Times | 1st | 2nd | 3rd | 4th | … | 14th | 15th | 16th | 17th |
|---|---|---|---|---|---|---|---|---|---|
| Drone 1 | 8:00–8:05 | 8:05–8:08 | 8:08–8:18 | 8:18–8:27 | … | 11:30–11:35 | | | |
| … | … | … | … | … | … | … | … | … | … |
| Drone 21 | 8:00–8:02 | 8:02–8:08 | 8:08–8:18 | 8:18–8:25 | … | 11:22–11:26 | 11:26–11:31 | 11:31–11:36 | |
| … | … | … | … | … | … | … | … | … | … |
| Drone 34 | 8:00–8:04 | 8:04–8:10 | 8:10–8:18 | 8:18–8:22 | … | 11:14–11:21 | 11:21–11:26 | 11:26–11:30 | 11:30–11:35 |
| … | … | … | … | … | … | … | … | … | … |
| Drone 50 | 8:00–8:07 | 8:07–8:14 | 8:14–8:21 | 8:21–8:28 | … | 11:27–11:35 | | | |

**Table 6.** Drone Dispatch Flight Schedule (BN).

| Delivery Times | 1st | 2nd | 3rd | … | 12th | 13th | 14th | … | 17th | 18th |
|---|---|---|---|---|---|---|---|---|---|---|
| Drone 1 | 8:00–8:05 | 8:05–8:17 | 8:17–8:17 | … | 11:27–11:30 | 11:30–11:36 | | | | |
| … | … | … | … | … | … | … | … | … | … | … |
| Drone 25 | 8:00–8:15 | 8:15–8:20 | 8:20–8:23 | … | 11:34–11:47 | | | | | |
| … | … | … | … | … | … | … | … | … | … | … |
| Drone 39 | 8:00–8:15 | 8:15–8:27 | 8:27–8:37 | … | 10:48–10:59 | 10:59–11:02 | 11:02–11:09 | … | 11:21–11:30 | 11:30–11:41 |
| … | … | … | … | … | … | … | … | … | … | … |
| Drone 50 | 8:00–8:05 | 8:05–8:17 | 8:17–8:22 | … | 11:11–11:19 | 11:19–11:29 | 11:29–11:35 | | | |

From Table 5, it can be seen that, during the RN route, drone 34 has the highest number of deliveries, at 17. Although drone 21 has been delivered 16 times, the delivery was completed the latest. It is worth noting that drone 21 needs to be charged at this time, so the total delivery time is 5 h and 36 min. From Table 6, it can be seen that, during the BN route, drone 39 has the highest number of deliveries, at 18. Although drone number 25 has only been delivered 12 times, the delivery was completed the latest. It is worth noting that drone 25 needs to be charged at this time, so the total delivery time is 5 h and 47 min. The difference in the final delivery time for each drone is not significant, indicating that the delivery plan has a certain degree of fairness and rationality.

### 4.3. Analysis of Algorithm Comparison

In solving drone demand, there are numerous feasible solutions due to the large amount of sample data, and different algorithms can have different impacts on the results. In this paper, we use a controlled experiment method to analyze the effects of five algorithms: delivery distance first, delivery parcel volume first, random delivery, standard simulated annealing, and improved simulated annealing on the drone demand results. A total of 100 experiments were conducted separately, and each experiment was set for 10,000 iterations, of which the respective optimal solution was taken to represent the following results:

Random distribution was carried out a total of one hundred times, and the minimum value was taken as the optimal solution and included in Table 7. After the random distribution experiment was completed, the maximum and minimum values of the respective distribution costs of the RN and BN routes were counted with the mean and variance as shown in the following table:

**Table 7.** Results of drone scheduling schemes under different algorithms.

| Algorithm | RN/BN Delivery Cost | RN/BN Delivery Time |
|---|---|---|
| Delivery distance priority | 2207.14/1705.75 | 5 h 48 min 37 s/5 h 44 min 47 s |
| Delivery package volume priority | 2172.33/1737.61 | 5 h 37 min 33 s/5 h 47 min 32 s |
| Random delivery | 2236.30/1654.93 | 5 h 45 min 43 s/5 h 50 min 32 s |
| Standard simulated annealing | 2281.58/1713.23 | 5 h 41 min 5 s/5 h 46 min 44 s |
| Improved simulated annealing | 2089.11/1626.37 | 5 h 36 min 33 s/5 h 47 min 44 s |

It can be seen from Table 7 that the improved simulated annealing algorithm adopted in this paper can obtain a more ideal delivery cost and delivery time than the other four algorithms, showing the accuracy and applicability of the algorithm. As seen in Table 8, the BN route itself is superior to the RN route, illustrating the feasibility of the aerial network constructed through the buildings. The average value of the distribution cost by the RN and BN routes illustrates that a better solution can be obtained by the improved heuristic algorithm, and the solutions obtained by the improved simulated annealing are improved by 8.4% and 5.1%, respectively, compared to the results of the randomized test, which illustrates the effectiveness of the improved simulated annealing algorithm.

**Table 8.** Random delivery statistics results.

| | Maximum Value | Minimum Value | Average Value | Variance |
|---|---|---|---|---|
| RN | 2326.42 | 2236.30 | 2272.81 | 361.64 |
| BN | 1758.34 | 1654.93 | 1699.04 | 428.72 |

*4.4. Analysis of Algorithm Parameters*

When solving drone demand, drone speed, load, and working hours will affect drone demand, as will the number of batteries at the distribution point, the number of drones, and other parameters. In this paper, controlled experiments were carried out to analyze the influence of the number of backup batteries, working hours, and charging time on drone demand when the number of drones changes.

Keep the settings of the other parameters as shown in Tables 2 and 3 unchanged. The number of drones is set to be 20, 30, 40, 50, 60, 70, and 80, respectively. The number of backup batteries is 10, 15, 20, 25, and 30, respectively. Multiple controlled experiments were conducted under the same conditions as other parameter settings and the airspace environment. The delivery costs required for the two different routes in each case were obtained, and the results are shown in Figure 6. The left side represents the RN route, the right side represents the BN route, and the unfilled hollow circle indicates the delivery cost under this condition, while the filled circle indicates that the lowest delivery cost has been reached at this time.

It can be seen from Figure 6 that the delivery cost is related to the changes in the number of drones and the number of backup batteries. This cost generally shows a downward trend and then rises. When the number of drones is 70 and the number of backup batteries is 15, the delivery cost of the RN route is the lowest. The BN route has the lowest delivery cost when the number of drones is 70 and the number of backup batteries is 20. Therefore, these numbers of drones and backup batteries can be considered when the actual logistics drones are delivered.

Keep the other parameters as shown in Tables 2 and 3 unchanged. The number of drones is set to be 20, 30, 40, 50, 60, 70, and 80, respectively. The working hours are 10 min, 20 min, 30 min, and 40 min. Multiple controlled experiments were carried out under the same conditions as other parameter settings and the airspace environment. The delivery costs of the two different routes in each case were obtained, and the results are shown in Figure 7. The left side represents the RN route, the right side represents the BN route, and

the unfilled hollow circle indicates the delivery cost under this condition, while the filled circle indicates that the lowest delivery cost has been reached at this time.

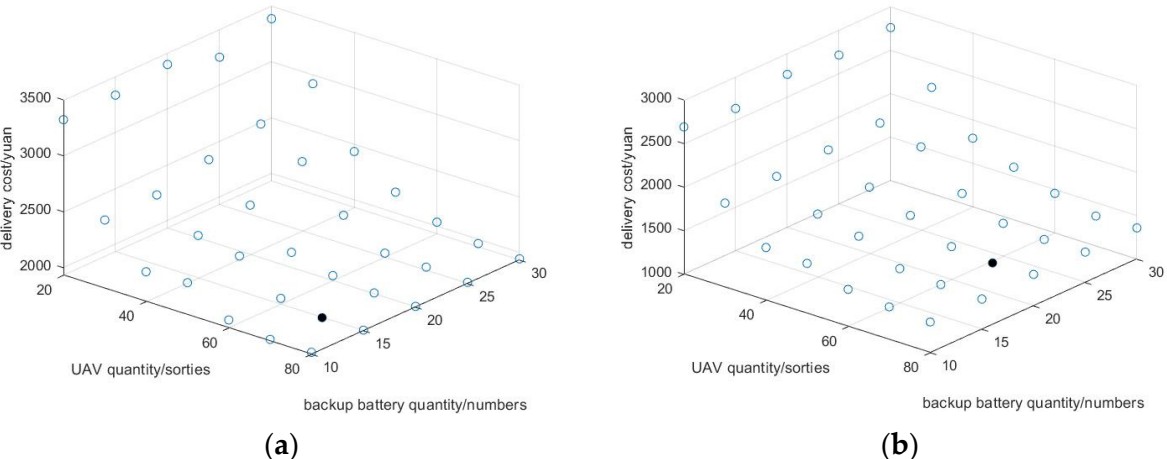

(**a**)

(**b**)

**Figure 6.** Schematic diagram of relationship between drone quantity/backup battery quantity and delivery cost, where (**a**) represents the RN route and (**b**) represents the BN route.

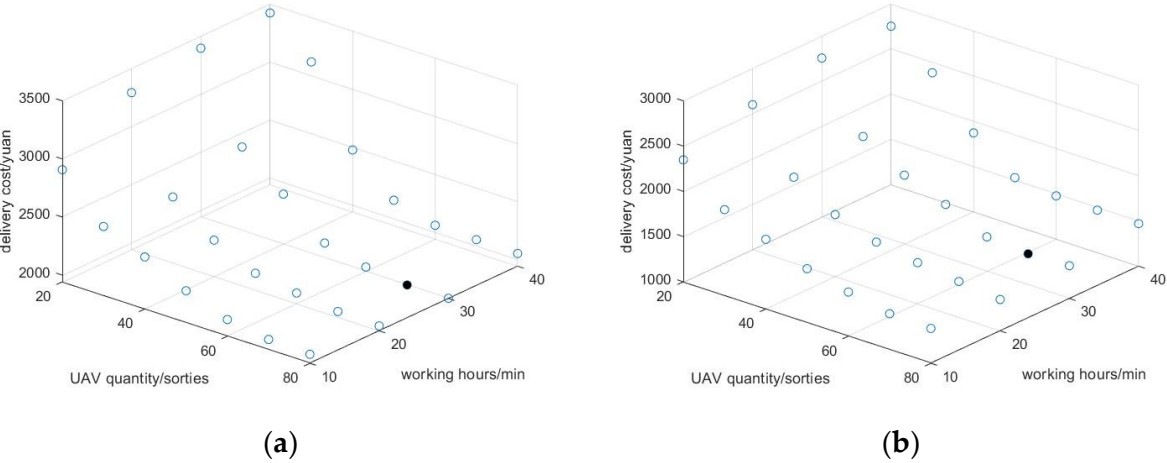

(**a**)

(**b**)

**Figure 7.** Schematic diagram of relationship between drone quantity/working hours and delivery cost, where (**a**) represents the RN route and (**b**) represents the BN route.

It can be seen from Figure 7 that the delivery cost is related to the changes in the number of drones and working hours. This cost generally shows a trend of decreasing and then rising. When the number of drones is 70 and the working time is 30 min, the delivery costs of both the RN route and the BN route are the lowest. Therefore, when the actual logistics drones are used for distribution, these results of drone numbers and working time can be considered.

Keep the other parameters as shown in Tables 2 and 3 unchanged. The number of drones is set to be 20, 30, 40, 50, 60, 70, and 80, respectively. The charging times are 1 h, 1.5 h, 2 h, and 2.5 h, respectively. Several controlled experiments were carried out under the same conditions as other parameter settings and the airspace environment. The delivery costs required under two different route situations were obtained, and the results are shown in Figure 8. The left side represents the RN route, the right side represents the BN route, and the unfilled hollow circle indicates the delivery cost under this condition, while the filled circle indicates that the lowest delivery cost has been reached at this time.

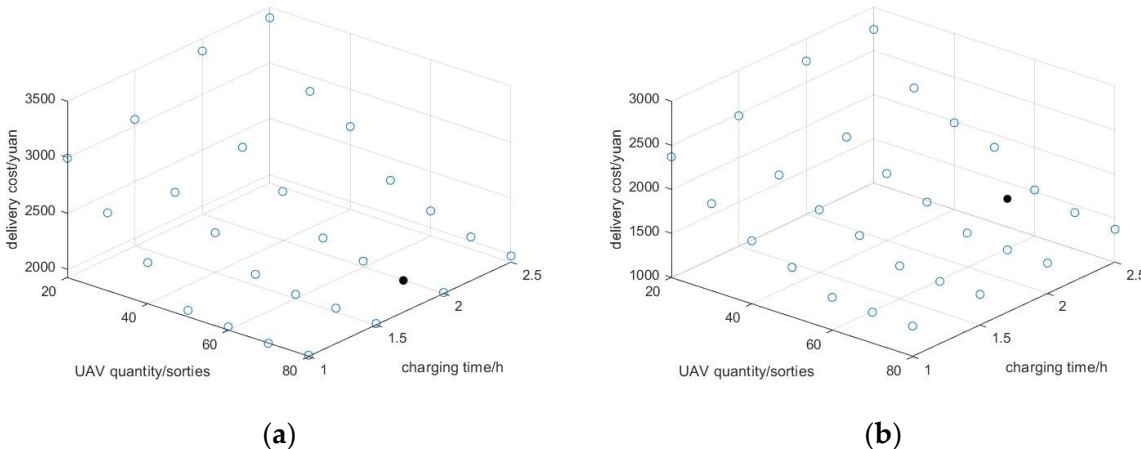

**Figure 8.** Schematic diagram of relationship between drone quantity/charging time and delivery cost, where (**a**) represents the RN route and (**b**) represents the BN route.

It can be seen from Figure 8 that the delivery cost is related to the changes in the number of drones and charging time. This cost generally shows a downward trend and then rises. When the number of drones is 70 and the charging time is 2 h, the RN route has the lowest delivery cost. When the number of drones is 70 and the charging time is 1.5 h, the delivery cost of the BN route is the lowest. Therefore, these numbers of drones and charging times can be considered when the actual logistics drones are delivered.

To sum up, in this case, the preferred number of drones is 70. In addition, depending on the choice of route (the RN route or the BN route), 15 or 20 backup batteries can be selected, the working time can both be 30 min, and the charging time can be 2 h or 1.5 h, respectively.

The above is a sensitivity analysis of several parameters. Through the analysis, it is found that the distribution demand is related to the number of drones, the number of backup batteries, the drone working hours, and the charging time. Combined with the delivery cost, it can be found that a higher number of backup batteries does not translate into a lower cost. After meeting the requirements for working hours within the scope of responsibility of the distribution point, increasing the working time of the drones cannot reduce the delivery cost. In addition, shorter charging times do not mean better results. With the increase in various parameters, the delivery cost generally tends to decrease first and then increase. Therefore, when purchasing drones, it is necessary to select the appropriate drones according to the distribution tasks of the distribution point.

## 5. Conclusions

Based on the improved simulated annealing algorithm and considering time cost, risk cost, maneuvering cost, and other related factors, this paper constructs a de-mand-based scheduling scheme model for urban low-altitude logistics drones. This model cannot only make the drone requirements reach the ideal stage but also calculate the drone requirements according to changes in relevant parameters.

This paper uses the RN/BN network to construct the scheduling scheme model and compare it with other algorithms. The experiments show that the model proposed in this paper outperforms other models in terms of distribution cost and distribution time and can better find out the optimal distribution route and optimal distribution cost. The improved simulated annealing in this paper improves the solution by 5~9% compared with the standard simulated annealing, which shows the rationality of the improved simulated annealing and also provides a reference basis for the logistics drone configuration in distribution centers and a reasonable scheduling method for future drone logistics, facilitating its normalization.

The drawback of this article is that it did not fully consider safety factors and did not dynamically consider factors such as the safety interval and collision risk of drones. Although the two routes in this article consider relevant factors in path planning, specifically within the route, only the ideal situation is considered in this article. In future research, we will attempt to study the impact of other complex parameters on logistics drones by including safety considerations.

At present, due to the lack of relevant data on urban low-altitude drone logistics distribution, actual drone distribution data are lacking. Particularly, there is little research on the air distribution network of urban low-altitude logistics, and the relevant data are insufficient. We can only assume the drone air distribution network through a few channels and simulate and estimate the drone distribution data. The next step will be to study the drone scheduling scheme combined with actual drone data and then verify the accuracy of the scheme.

**Author Contributions:** Conceptualization, H.Z.; Investigation, O.F.; Data curation, T.T.; Writing—original draft, S.W.; Visualization, Y.H.; Supervision, G.Z. All authors have read and agreed to the published version of the manuscript.

**Funding:** This research was funded by the National Natural Science Foundation of China (71971114) and the Fundamental Research Funds for the Central Universities (No. NQ2022012).

**Institutional Review Board Statement:** Not applicable.

**Informed Consent Statement:** Not applicable.

**Data Availability Statement:** Not applicable.

**Acknowledgments:** The authors acknowledge with thanks the support for this work by the College of Civil Aviation at Nanjing University of Aeronautics and Astronautics.

**Conflicts of Interest:** The authors declare no conflict of interest.

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
