# Peer review of "Research on Demand-Based Scheduling Scheme of Urban Low-Altitude Logistics UAVs"

_applsci, doi:10.3390/app13095370_

Round 1

Reviewer 1 Report

Flaws:

I have reviewed the paper and I have found multiple concerns. Before listing them, I would also need to mention that the paper is badly written and not well structured. The paper needs a lot of editing in these aspects.

1. Cited literature is incomplete. I think the authors only focused on research in China because there are several papers on the topic not included for instance Chauhan et al 2023, Glick et al 2022, DellAmico et al (2021). 

2. There are assumptions that are very strong and do not correlate with the literature. For instance, the UAVs are affected by the weight as battery which reduces the number of possible deliveries if packages are heavy. Similarly with the effect of height. At the end, with these strong assumptions the model is oversimplified. 

3. The authors need to provide a better explanation of the simulated annealing method. Why is simulation annealing better than other metaheuristics? Simulated annealing usually needs to deal with multiple local optimums. For this reasons sometimes it needs to be blended with other methods. Besides, I do not see the improved part in the simulation annealing. To me it seems the standard procedure.

4. Also as simulated annealing is very sensitive to its parameters, can the authors provide an explanation of how they were chosen? (In particular Table 

5. A standard method for comparing methods is to create instances and run multiple experiments. Then proceed with a statistical test. If you use a sample of size n=1 there is no guarantee that your claimed results are better. For instance, in Table 5 the results of RB for distance delivery and simulated annealing are very similar. Maybe if you multiple experiments you will end up with no statistical differences.

REFERENCES

Rajesh Chauhan, D., Unnikrishnan, A., Figliozzi, M. A., & Boyles, S. D. (2023). Robust Multi-Period Maximum Coverage Drone Facility Location Problem Considering Coverage Reliability. Transportation Research Record2677(2), 98–114. https://doi.org/10.1177/03611981221087240

Glick, T. B., Figliozzi, M. A., & Unnikrishnan, A. (2022). Case Study of Drone Delivery Reliability for Time-Sensitive Medical Supplies With Stochastic Demand and Meteorological Conditions. Transportation Research Record2676(1), 242–255. https://doi.org/10.1177/03611981211036685

Dell’Amico, M., Montemanni, R. & Novellani, S. Drone-assisted deliveries: new formulations for the flying sidekick traveling salesman problem. Optim Lett 15, 1617–1648 (2021). https://doi.org/10.1007/s11590-019-01492-z

Reviewer 2 Report

A brief summary 

The article focuses on UAV logistic system optimization to minimize the total delivery cost and delivery time function. The used model is based on Shanghai’s low air and ground traffic network, which is one of the biggest strengths of this research. 

Three optimization methods are used, including 3 dispatching rules (delivery distance, package volume, random) and simulated annealing with default parameters of UAV (speed, range, cargo capacity), logistic system capacity (number of batteries, UAVs, working hours, charging duration) and risk costs coefficients.

Logistic system optimization of delivery cost on previously mentioned networks is performed in the second part of the article. The major factor is the number of UAVs and secondary factors are the number of UAVs backup batteries, working hours duration, and charging time. 

Comments

First and foremost, I think the paper fits the journal’s scope and special issue topic. It covers a very interesting field of UAV logistics. The great benefit of this paper is the real-world model.

Still, I do not recommend publishing this paper without improving these major issues:

Scientific content

S 1 Literature review – is limited and includes self-citations of non-English references.

S 1.1 State-of-the-art literature references in UAS scheduling and routing are missing. Numerous sources focus on UAV's performance (e.g. load, charging) and environmental constraints (e.g. wind, area legislation). The article points out that such research is limited. I disagree.

S 1.2 Article references sources [10-13], are unavailable in English. Moreover, it seems it references the author’s (Zang,H.) publications [10,11]. Please substitute them with relevant sources in English or remove them while describing referenced content in this article.

S 1.3 One of the biggest constraints is usually legal restrictions for UAVs in the terms of restricted space where they can be operated. It would improve the paper's relevance if such criteria are described.

S 2 Scheduling-routing model - its solution and schedule presentation are missing.

The major part of model optimization – schedule and its creation is not presented despite being defined as one of the article outcomes. See “Planning ideal delivery route” (2nd step in chapter 1.1 – lines 91-92).

It is limited to one reference [10] which is in the Chinese, sentence in chapter “3.2 Analysis of calculation results” (lines 268-270) “use Python programming to calculate the future UAV scheduling scheme of the distribution point „ and block in Figure 1 “Dynamic allocation - Calculate feasible delivery scheme” 

The authors should make clear:

S 2.1 How tij is calculated (line 114)?

S 2.2 How (based on what – what are the rules) height hm (7) is set (line 135)? 

The practical model is stating that only Min and Max values are set for 100m departure and 150m for return. What happens when 2 UAVs are returning/departing from opposite directions so they are on the same level going one against each other?

S 2.3 How (based on what – what are the rules) speed vm (6) is set (line 140). Authors are mentioning that delivery time (speed) is not affected by the weight of the delivered object. What then influences the speed? Is vmin for loaded and vmax for unloaded?

S 2.4 What are other constraints on routes (minimal distance between UAVs etc.)? How is the path generated so UAV will not collide with other UAVs etc.?

S 2.5 How are UAVs scheduled on their routes? A mathematical model, scheduling “pseudo” algorithm, or process diagram as in Figure 1, is missing.

S 2.6 Representation of at least one solution schedule (from table 5 pp 11), e.g. the one authors are showing at reference [10] (fig 3).

S 3 Model constraints are not clearly described

S 3.1 It is not clear how “(2) Airspace constraint” values variables (see my comments S 2.3, S 2.4) are set/decided.

S 3.2 Time cost constraints (equations (8-12) lines 141 to 173) are not constraints. 

There are no limits to them. It seems like it is the elaboration of multi-objective function costs. Please make clear how these costs are limited or move this part from the model constraint to the objective function section (sub-chapter 1.3.2). 

S 4 Multicriteria optimization, Z criteria are unclear – hard to understand, as chapter 1.3.2 is poorly organized.

S 4.1 It is not clear what Z2 represents. It seems like it’s a risk that UAVs have to wait because spare batteries are unavailable and UAVs are out of power. The main reason is that x and y in equation (14) are described in (4) and (5). That problem applies to Z1 and Z3 as well.

S 4.2 Z2 criteria, as well as other equations (9),(13) (14), are missing an explanation of what A and J are. Probably A is the maximal number of batteries (Table 2 defines it as N) and J is the maximum number of demand points. 

Regarding points 4.1 and 4.2

Please consider reorganization of chapters 1.3.1 and 1.3.2 as follows:

  1. Define the whole objective function as in (16) and explain function – motivation of each Z

  2. Define Z1 as in (13) and elaborate the meaning of variables and periods equations (8) etc.

  3. Define Z2 (14) and the following variables as in (9-11) etc.

  4. Define Z3 (15) and its variables (12) etc.

S 4.3 Scope of the cost function is limited -

as it is based only on the operation costs, not purchasing cost of equipment (UAVs and batteries).

That can influence conclusions based on experiments in chapter “3.4 Analysis of algorithm parameter”. 

That can lead to (maybe) wrong assumptions about the influence of the number of batteries (15 or 20 at line 344) and UAVs, as is concluded in lines 349-350, “it can be found that the number of backup batteries is not the more, the better”.

However, I don’t think that the cost function has to be changed. It could help just to explain more clearly the motivation behind its criteria or to address its limits.

S 5 Improved simulated annealing algorithm 

Model parameters optimized by Simulated Annealing are not defined and explained.

S 6 Experiment setup and result evaluation- parameters and statistical evaluation are unclear/missing.

S 6.1 It is unclear which method is used to obtain the results shown in Table 4. It seems that it is Improved Simulated Annealing from Table 5. Why are these results presented two times?

S 6.2 Assuming Random delivery and Simulated annealing (Table 5) are stochastic methods. In that case, I’m missing the following information:

6.2.1 How many experiments were done? The article is mentioning “multiple experiments were done”

6.2.2 What are the statistical properties of stochastic experiments (average, min/max, deviation, etc.)?

S 6.3 It is not possible to state that dynamic allocation improved Simulated Annealing if Improved Simulated Annealing is not compared with the non-improved. 

Please add results of simple Simulated Annealing together with sound statistical comparison e.g. testing for statistically significant differences.

Formal Content

F1 Literature numbers referenced in the text are all in uppercase style except the one on line 70 [10,11]. Please format referencing so it is not in upper case - aligned with the text.

F2 All variables are in uppercase as well. Please align all variables.

F3 All equations, except (17) have numbering in lowercase. Please align all equations with their numbering.

F4 Please don't use numbers in round brackets “(1)” as a tool to structure text as it is dedicated to equation numbering only.

F5 I recommend adding one more column (Table 2) separately for units of set parameters. Separator “/” can lead to wrong assumptions that it is part of the variable definition.

Conclusion

I recommend publishing this paper only after:

  • The literature review is improved (S1 comments)

  • Scheduling - Routing algorithm is presented and an example of a solution is shown (S2).

  • Variables and models are clearly described (S2 and S3)

  • The objective function section is reorganized so it is easy to comprehend (S4)

  • Parameters optimized by proposed simulated annealing are defined. (S5)

  • Statistical properties of the Experiment are presented and Improved Simulated Annealing is compared to the non-improved version (S6).

Reviewer 3 Report

The manuscript entitled “Research on demand-based scheduling scheme of urban low altitude logistics UAVs” seems to be acceptable to be considered for publication in applied sciences, but it would be better if some issues be regarded in the revision.
1. Since this paper is not a literature review, in lieu of referring to a vast number of previous works, which have been surveyed by other researchers, mention the latest related works.

Wu, Yu, Kin Huat Low, and Xinting Hu. "Trajectory-based flight scheduling for AirMetro in urban environments by conflict resolution." Transportation Research Part C: Emerging Technologies 131 (2021): 103355.

Eslamipoor, R. (2022). An optimization model for green supply chain by regarding emission tax rate in incongruous vehicles. Modeling Earth Systems and Environment, 1-12.

A two-stage stochastic planning model for locating product collection centers in green logistics networks." Cleaner Logistics and Supply Chain 6 (2023): 100091.

She, Ruifeng, and Yanfeng Ouyang. "Efficiency of UAV-based last-mile delivery under congestion in low-altitude air." Transportation Research Part C: Emerging Technologies 122 (2021): 102878.
2. The transitions from topic to topic in the paper seem to be a little sudden. In other words, while reading about a topic, the text suddenly starts to mention something quite different. It is suggested to smooth these transitions from topic to topic where possible.
3. Please indicate the contributions in more detail, specifically in comparison with the latest research papers.
4. Above all, please polish the title to be in line with the contents of this manuscript.

Reviewer 4 Report

1. There have been many studies on logistics UAVs both at home and abroad - at least some references to prove it would be useful.

2. The abbreviation SUV was used 72 times (!) in the text.

3. The assumptions should be a bit better described. Only these assumptions apply in the paper?

4. The list of references is far too short. The paper should be better based on a literature review.

5. The presentation of the results should be slightly better, it is also about the readability of the graphics.

Reviewer 5 Report

This paper proposes a demand-based scheduling scheme for urban low altitude logistics unmanned aerial vehicles (UAVs) that considers various constraints such as UAV performance, airspace, and distribution. The scheme is verified using actual data and airspace constraints from Shanghai and is shown to outperform other forecasting models in terms of delivery cost and time, while also providing flexibility in calculating the optimal scheduling scheme under multiple parameters. There are various reasons that make this article ineligible for acceptance, which will be discussed in detail. Despite the effort put into writing this piece, it falls short in meeting the criteria required for acceptance. Therefore, further comments are necessary to consider.

1.     One major concern that I have with this article is its apparent lack of contribution to the field. The paper fails to demonstrate any significant novelty or innovation in its approach, leaving me uncertain as to its original contribution to the existing body of research. In my opinion, the article fails to present any unique or innovative methodology or approach, thus failing to make a significant advancement in the current state of research within this field.

2.     A comprehensive literature review is essential for contextualizing the research problem, identifying gaps in previous research, and justifying the need for the current study. It helps to establish the significance and novelty of the study and provides a theoretical foundation for the research. In this article, the literature review is brief and superficial, leaving the reader with a sense of incompleteness and ambiguity. Furthermore, the low number of references cited in the paper suggests that the authors did not conduct a thorough review of the relevant literature. This can undermine the credibility and validity of the research findings and limit the scope of the study.

3.     Indices, sets, parameters, and decision variables are crucial elements of any mathematical model, and they must be appropriately defined and presented in the paper. These elements define the scope and complexity of the problem, and they provide a framework for developing the model. In this article, these elements are not presented in a clear and concise manner, making it challenging for the reader to understand the problem's scope and the model's underlying assumptions. This can lead to misunderstandings and misinterpretations, which can undermine the credibility and validity of the research findings. To address this issue, the authors should revise the article to provide a more comprehensive and clear presentation of the indices, sets, parameters, and decision variables used in the modeling process.

4.     According to the authors, the model put forth in the paper is a single-objective model. However, it is apparent that the original model was three-objective, which was transformed into a single-objective model by incorporating coefficients to convert all objectives into cost. Nevertheless, the paper lacks an adequate explanation on the derivation of these coefficients, as well as a sensitivity analysis of the model based on these coefficients, thus underscoring a neglected aspect of the study.

5.     The algorithm put forth by the authors exhibits a deficiency in terms of originality. The article must explicitly acknowledge this shortcoming and take necessary steps to rectify it. It is crucial for academic research to contribute to the existing knowledge base by introducing novel and innovative ideas. Therefore, the absence of novelty in the proposed algorithm is a matter of concern that requires prompt attention. The authors should conduct a thorough literature review to ensure that their algorithm is distinct from prior works and identify areas where they can introduce novel contributions. By doing so, they can add value to the field and enhance the credibility of their research.

Round 2

Reviewer 1 Report

The paper is better presented. However, I am still concerned about the experiment (and therefore the results). To better claim the results a proper design of experiments is needed and statistical test of the results. I previously suggested that a minimum expected is a hypothesis test of the results shown in Table  8. That is, run (random) experiments and then compare the means through an Anova.

Reviewer 2 Report

Dear authors,

your paper was significantly improved, however, it is still not enough for publishing.

You reflected all of my comments in your author response but did not improve them - reference them in your paper. For some of them, you did not address them in a way that would be understandable to me. 

I strongly believe that you accidentally uploaded the wrong revised file (the new file is different from the original one) as in a lot of cases you address that you made changes, however, it is not reflected in the paper.

An example can be my formal comments which u addressed in the letter but there is no change in the paper.

I still do not recommend publishing this paper without improving the points enclosed in the attached pdf file. 

Im rating the current state as " Reconsider after a major revision (Control missing in some experiments" 

The description of the UAV's scheduling/routing/path planning algorithm is still missing and results are not evaluated in the usual scientific dept (see comments in attached file).

Reviewer 4 Report

I still find the review insufficient.
1. Please organize the literature, group it into topics. It's not enough to write that someone published about it and someone else about something else. Because what does it mean? Please read the state of the art articles. See what the overview looks like there, a look from the most important pages is shown.
2. There are still a huge number of "UAV" repetitions. I will not allow the article in this form to be published. Edit the text properly.
3. I want to see the calculation assumptions and methods along with the justification.
4. The presentation of the results should be slightly better, it is also about the readability of the graphics. Not much has changed here, not enough for me to accept.

Reviewer 5 Report

-
